# Robust Recursive Partitioning for Heterogeneous Treatment Effects with Uncertainty Quantification

**Hyun-Suk Lee**[*]
Sejong University
hyunsuk@sejong.ac.kr

**Yao Zhang**[*]
University of Cambridge
yz555@cam.ac.uk

**William R. Zame**
UCLA
zame@econ.ucla.edu

**Cong Shen**
University of Virginia
cong@virginia.edu

**Jang-Won Lee**
Yonsei University
jangwon@yonsei.ac.kr

**Mihaela van der Schaar**
University of Cambridge
UCLA
The Alan Turing Institute
mv472@cam.ac.uk

## Abstract

Subgroup analysis of treatment effects plays an important role in applications from medicine to public policy to recommender systems. It allows physicians (for example) to identify groups of patients for whom a given drug or treatment is likely to be effective and groups of patients for which it is not. Most of the current methods of subgroup analysis begin with a particular algorithm for estimating individualized treatment effects (ITE) and identify subgroups by maximizing the differences across subgroups of the average treatment effect in each subgroup. These approaches have several weaknesses: they rely on a particular algorithm for estimating ITE, they ignore (in)homogeneity within identified subgroups, and they do not produce good confidence estimates. This paper develops a new method for subgroup analysis, R2P, that addresses all these weaknesses. R2P uses an arbitrary, exogenously prescribed algorithm for estimating ITE and quantifies the uncertainty of the ITE estimation, using a construction that is more robust than other methods. Experiments using synthetic and semi-synthetic datasets (based on real data) demonstrate that R2P constructs partitions that are simultaneously more homogeneous within groups and more heterogeneous across groups than the partitions produced by other methods. Moreover, because R2P can employ any ITE estimator, it also produces much narrower confidence intervals with a prescribed coverage guarantee than other methods.

## 1   Introduction

The understanding of treatment effects plays an important role – especially in shaping interventions and treatments – in areas from clinical trials [1, 2] to recommender systems [3] to public policy [4]. In many settings, the relevant population is diverse, and different parts of the population display different reactions to treatment. In such settings, *heterogeneous treatment effect (HTE) analysis* – also called *subgroup analysis*– is used to find subgroups consisting of subjects who have similar covariates and display similar treatment responses [5, 6]. The identification of subgroups is informative of itself; it also improves the interpretation of treatment effects across the entire population and makes it possible to develop more effective interventions and treatments and to improve the design of further experiments. In a clinical trial, for example, HTE analysis can identify subgroups of the population

---

[*]Equal contribution

for which the studied treatment is effective, even when it is found to be ineffective for the population in general [7].

To identify subjects who have similar covariates and display similar treatment responses, it is necessary to create reliable estimates of the treatment responses of individual subjects; i.e. of *individualized treatment effects (ITE)*. The state-of-the-art work on HTE proceeds by simultaneously estimating ITE and recursively partitioning the subject population [8–11]. In these HTE methods, the criterion for partitioning is maximizing the heterogeneity of treatment effects *across* subgroups, using a sample mean estimator, under the assumption that treatment effects are homogeneous *within* subgroups. In particular, the population (or any previously identified subgroup) would be partitioned into two subgroups provided that the sample means of these subgroups are sufficiently different, ignoring the possibility that treatment effects might be very heterogeneous within the groups identified. Put differently, these methods focus on inter-group heterogeneity but ignore intra-group heterogeneity.

An important problem with this approach is that, because it relies solely on inter-group heterogeneity based on sample means, it may lead to *false discovery*. To illustrate, consider the toy example depicted in Fig. 1. In this example, the true treatment effect (shown on the vertical axis) was generated by iid random draws from the normal distribution having mean 0.0 and standard deviation 0.1. In truth, the treatment under consideration is in fact *totally ineffective and innocuous*; on average, it has *no effect at all* and the treatment effects are entirely uncorrelated with the single covariate (shown on the horizontal axis). However if the observed data – the realization of the random draws – happens to be the one shown in Fig. 1, standard methods will typically partition the population as shown in

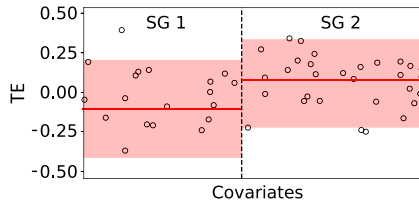

Figure 1: A toy example with two subgroups identified by HTE method in [10]. The solid red line shows the ITE estimation and their 95% confidence interval is filled in red.

the figure, thereby "discovering" a segment of the population for whom the treatment is effective and a complementary segment where the treatment is dangerous. Obviously, decisions based on such false discovery are useless – or worse. Note that this false discovery occurs because, although the outcome variations between the two groups are indeed substantially different, the outcome variations within each group are just as different – but the latter variation is entirely ignored in the creation of subgroups.

This paper proposes a robust recursive partitioning (R2P)[2] method that avoids such false discovery. R2P has several distinctive characteristics.

- *R2P discovers interpretable subgroups in a way that is not tied to any* particular *ITE estimator.* This is in sharp contrast with previous methods [8–12], each of which relies on a *specific* ITE estimator. R2P can leverage *any* ITE estimator for subgroup analysis, e.g. an ITE estimator based on Random Forest [13], or on multi-task Gaussian processes [14] or on deep neural networks [15–17]. This flexibility is important because no one ITE estimator is consistently the best in all different settings [18]. Furthermore, these ITE estimators are non-interpretable black-box models. R2P divides units into subgroups with respect to an interpretable tree-structure, and provides subgroup coverage guarantees for the ITE estimates in each subgroup. R2P enables these ITE estimators to produce trustworthy and interpretable ITE estimates in practice.

- *R2P makes a conscious effort to guarantee homogeneity of treatment effects within each of the subgroups while maximizing heterogeneity across subgroups.* This is also different from previous methods, e.g., [8, 10] where variation within the subgroup is largely ignored.

- *R2P produces* confidence guarantees *with narrower confidence intervals than previous methods.* It accomplishes this by using methods of conformal prediction [19] that produce valid confidence intervals. Quantifying the uncertainty allows R2P to employ a novel criterion we call *confident homogeneity* in order to create partitions that take account of both heterogeneity across groups and homogeneity within groups.

These characteristics make R2P both more reliable and more informative than the existing methods for subgroup analysis. Extensive experiments using synthetic and semi-synthetic datasets (based on real-world data) demonstrate that R2P outperforms state-of-the-art methods by more robustly identifying subgroups while providing much narrower confidence intervals.

## 2 Robust Recursive Partitioning with Uncertainty Quantification

To highlight the core design principles, we begin by introducing robust recursive partitioning (R2P) in the regression setting; we extend to the more complicated HTE setting in the next section.

### 2.1 Preliminaries

We consider a standard regression problem with a $d$-dimensional covariate (input) space $\mathcal{X} \subseteq \mathbb{R}^d$ and a outcome space $\mathcal{Y} \subseteq \mathbb{R}$. We are given a dataset $\mathcal{D} = \{(\mathbf{x}_i, y_i)\}_{i=1}^n$, where, for the $i$-th sample, $\mathbf{x}_i \in \mathcal{X}$ is the vector of input covariates and $y_i \in \mathcal{Y}$ is the outcome. We assume that samples are independently drawn from an unknown distribution $\mathcal{P}_{X,Y}$ on $\mathcal{X} \times \mathcal{Y}$. We are interested in estimating $\mu(\mathbf{x}) = \mathbb{E}[Y|X = \mathbf{x}]$, which is the mean outcome conditional on $\mathbf{x}$. We denote the estimator by $\hat{\mu} : \mathcal{X} \to \mathcal{Y}$; $\hat{\mu}$ predicts an outcome $\hat{y} = \hat{\mu}(\mathbf{x})$ on the basis of the covariate information $\mathbf{x}$. To quantify the uncertainty in the prediction, we apply the method of split conformal regression (SCR) [19] to construct a confidence interval $\hat{C}$ that satisfies the rigorous frequentist guarantee in the finite-sample regime. (To the best of our knowledge, SCR is the simplest method that achieves this guarantee.)

In SCR, we take as given a *miscoverage rate* $\alpha \in (0, 1)$. We split the samples in $\mathcal{D}$ into a training set $\mathcal{I}_1$ and a validation set $\mathcal{I}_2$ that are disjoint and have the same size. We train the estimator $\hat{\mu}^{\mathcal{I}_1}$ on $\mathcal{I}_1$ and compute the residual of $\hat{\mu}^{\mathcal{I}_1}$ on each sample in $\mathcal{I}_2$. For a testing sample $\mathbf{x}$ the confidence interval is given by

$$\hat{C}^{\mathcal{I}_1, \mathcal{I}_2}(\mathbf{x}) = \left[\hat{\mu}^{\mathrm{lo}}(\mathbf{x}), \hat{\mu}^{\mathrm{up}}(\mathbf{x})\right] = \left[\hat{\mu}^{\mathcal{I}_1}(\mathbf{x}) - \hat{Q}_{1-\alpha}^{\mathcal{I}_2}, \hat{\mu}^{\mathcal{I}_1}(\mathbf{x}) + \hat{Q}_{1-\alpha}^{\mathcal{I}_2}\right], \tag{1}$$

where $\hat{Q}_{1-\alpha}^{\mathcal{I}_2}$ is defined to be the $(1 - \alpha)(1 + 1/|\mathcal{I}_2|)$-th quantile of the set of residuals $\{|y_i - \hat{\mu}^{\mathcal{I}_1}(\mathbf{x}_i)|\}_{i \in \mathcal{I}_2}$. Assuming that the training and testing samples are drawn exchangeably from $\mathcal{P}_{X,Y}$, the confidence intervals defined in (1) satisfy the coverage guarantee $\mathbb{P}[y \in \hat{C}^{\mathcal{I}_1, \mathcal{I}_2}] \geq 1 - \alpha$ [19].[3]

To illustrate, assume the miscoverage rate $\alpha$ is 0.05 and we are given 1000 testing samples. SCR prescribes a confidence interval for each sample in such a way that for at least 950 samples the prediction is within the prescribed confidence interval. (We often say the sample is *covered*.) However this coverage guarantee is marginal over the entire covariate space. If we perform a subgroup analysis that partitions the covariate space $\mathcal{X}$ into subgroups $\mathcal{X}_1, \mathcal{X}_2$ in such a way that $\mathcal{X}_1$ has 800 samples and $\mathcal{X}_2$ has 200 samples, it might be the case that 790 samples in $\mathcal{X}_1$ are covered but only 160 samples in $\mathcal{X}_2$ are covered. In this case, 80% of the samples in $\mathcal{X}_2$ would be covered. It seems obvious that such a situation is undesirable for subgroup analysis; we want to achieve the prescribed coverage rate for *each* subgroup, not just for the population as a whole. R2P overcomes this problem.

We begin by discussing how we use confidence intervals to quantify outcome homogeneity within a subgroup. We then introduce our space partitioning algorithm and provide the required theoretical guarantee of subgroup coverage.

### 2.2 Partitioning for Robust Heterogeneity Analysis

Let $\Pi$ be a partition of the covariate space $\mathcal{X}$ into (disjoint) subgroups. Write $|\Pi|$ for the number of subgroups in the partition, $l_j$ for an element of $\Pi$ and $l(\mathbf{x}; \Pi)$ for the subgroup that contains the sample $\mathbf{x}$. Write $\mathcal{D}_l = \{(\mathbf{x}_i, y_i) \in \mathcal{D}|\mathbf{x}_i \in l\}$ for the samples whose covariates belong to the subgroup $l$. Note that when we restrict to covariates in the subgroup $l$, the samples are drawn from the truncated distribution $\mathcal{P}_{X,Y}^l$ which is the distribution conditional on the requirement that the vector of covariates of samples lie in the subgroup $l$.

We evaluate homogeneity within the subgroup $l$ by the concentration of outcome values for covariate vectors in the subgroup $l$. To do this, we apply SCR to the samples in $\mathcal{D}_l$ by splitting it into two sets, $\mathcal{I}_1^l$ and $\mathcal{I}_2^l$. Write $\hat{\mu}_l(\mathbf{x})$ denote the mean outcome model trained on $\mathcal{I}_1^l$. As in (1), we obtain the confidence interval $\hat{C}_l(\mathbf{x})$ for subgroup $l$ by setting the upper and lower endpoints to be $\hat{\mu}_l^{\mathrm{up}}(\mathbf{x}) = \hat{\mu}_l(\mathbf{x}) + \hat{Q}_{1-\alpha}^{\mathcal{I}_2^l}$ and $\hat{\mu}_l^{\mathrm{lo}}(\mathbf{x}) = \hat{\mu}_l(\mathbf{x}) - \hat{Q}_{1-\alpha}^{\mathcal{I}_2^l}$, respectively. (To avoid notational complications, omit reference to the subsets $\mathcal{I}_1^l$ and $\mathcal{I}_2^l$ hereafter; this should not cause confusion. Throughout, we follow the convention that the confidence bound have been computed on the basis of the split.) To estimate the center of the subgroup $l$, we use the average outcome $\hat{\mu}_{l,\mathrm{mean}} = \mathbb{E}[\hat{\mu}_l(\mathbf{x})]$.

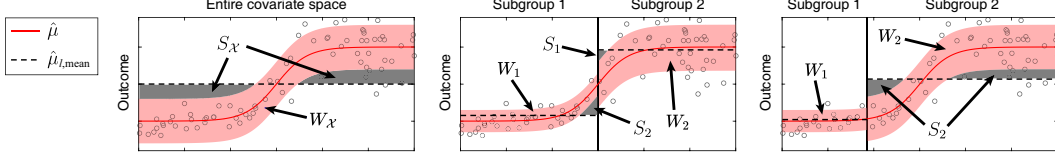

Figure 2: **Illustration of partitioning and impurity of the confident homogeneity.** The regions shaded in red and gray (roughly) represent $W_l$ and $S_l$, respectively. Partitioning the heterogeneous covariate space (left panel) reduces its impurity of the confident homogeneity. The partition with the smaller impurity (middle panel) makes the heterogeneity across subgroups and the homogeneity within subgroups both stronger compared to others with the larger impurity (e.g., right panel).

We define the *expected absolute deviation* in group $l$ to be $S_l = \mathbb{E}[v_l(\mathbf{x})]$, where

$$v_l(\mathbf{x}) = \left(\hat{\mu}_{l,\text{mean}} - \hat{\mu}_l^{\text{up}}(\mathbf{x})\right) \mathbb{I}\left[\hat{\mu}_{l,\text{mean}} > \hat{\mu}_l^{\text{up}}(\mathbf{x})\right] + \left(\hat{\mu}_l^{\text{lo}}(\mathbf{x}) - \hat{\mu}_{l,\text{mean}}\right) \mathbb{I}\left[\hat{\mu}_{l,\text{mean}} < \hat{\mu}_l^{\text{lo}}(\mathbf{x})\right]. \quad (2)$$

By definition, $\hat{\mu}_l^{\text{up}}(\mathbf{x})$ is larger than $\hat{\mu}_l^{\text{lo}}(\mathbf{x})$ provided the residual quantile $\hat{Q}_{1-\alpha} > 0$. When the first indicator function is one, i.e. the average outcome (the group center) $\hat{\mu}_{l,\text{mean}}$ is larger than the upper bound $\hat{\mu}_l^{\text{up}}(\mathbf{x})$ at $\mathbf{x}$, we are confident that the outcome value at $\mathbf{x}$ is smaller than the group center (and perhaps smaller than the outcome value for many covariate vectors in $l$). Similarly, when the second indicator function is one, we are certain that the outcome value at $\mathbf{x}$ is larger than the group center (and perhaps larger than the outcome value for many covariate vectors in $l$). It is worth noting that when $\hat{C}_l(\mathbf{x}) = \left[\hat{\mu}_l^{\text{lo}}(\mathbf{x}), \hat{\mu}_l^{\text{up}}(\mathbf{x})\right]$ contains the center $\hat{\mu}_{l,\text{mean}}$, both indicator functions are zero and $v_l(\mathbf{x}) = 0$. The quantity $v_l(\mathbf{x})$ evaluates the homogeneity in subgroup $l$ on the basis of the confidence interval for each $\mathbf{x}$ in $l$. (It is more conservative than the mean discrepancy $|\hat{\mu}_l(\mathbf{x}) - \hat{\mu}_{l,\text{mean}}|$ for partitioning, and hence provides greater protection against false discovery because of uncertainty.)

However, minimizing $S_l$ is not enough to maximize subgroup homogeneity. If the intervals $\hat{C}_l(\mathbf{x})$ for all $\mathbf{x} \in l$ are very wide and contain the average outcome $\hat{\mu}_{l,\text{mean}}$, homogeneity can be very low even though $S_l = \mathbb{E}[v_l(\mathbf{x})]$ is zero. To resolve this issue, when partitioning the covariate space we jointly minimize $S_l$ and the expected confidence interval width $W_l = \mathbb{E}\left[|\hat{C}_l(\mathbf{x})|\right]$. We formalize the robust partitioning problem as

$$\underset{\Pi}{\text{minimize}} \sum_{l \in \Pi} \lambda W_l + (1 - \lambda) S_l, \quad (3)$$

where $\lambda \in [0, 1]$ is a hyperparameter that balances the impact of $W_l$ and $S_l$. We call the weighted sum, $\lambda W_l + (1 - \lambda) S_l$, the impurity of the *confident homogeneity* for subgroup $l$. Fig. 2 illustrates how minimizing the impurity of the confident homogeneity improves both homogeneity within subgroups and heterogeneity across subgroups. There may be more than one partition that achieves the minimum; because a larger number of subgroups is harder to interpret, we will choose a minimizer with the smallest number of subgroups.

## 2.3 Robust Recursive Partitioning Method with Confident Homogeneity

We can now describe our robust recursive partitioning (R2P) method for solving the optimization problem (3). We begin with the trivial partition $\Pi = \{\mathcal{X}\}$. We denote the set of subgroups whose objectives in (3) can be potentially improved by $\Pi_c$. In the initialization step, we set $\Pi_c = \Pi$ and apply the SCR on $\mathcal{D}$ to obtain the confidence intervals in (1). Based on these intervals, we compute $\hat{W}_\mathcal{X}$ and $\hat{S}_\mathcal{X}$. More generally, using the intervals for subgroup $l$, we can estimate $W_l$ and $S_l$ by

$$\hat{W}_l = \frac{1}{N_2^l} \sum_{i \in \mathcal{I}_2^l} |\hat{C}_l(\mathbf{x}_i)| \text{ and } \hat{S}_l = \frac{1}{N_2^l} \sum_{i \in \mathcal{I}_2^l} v_l(\mathbf{x}_i),$$

respectively, where $N_2^l$ is the number of samples in the validation set $\mathcal{I}_2^l$.

After initialization, we recursively partition the covariate space by splitting the subgroups in $\Pi_c$ with respect to the criterion in (3). To split each subgroup $l \in \Pi_c$, we first consider the two disjoint subsets from subgroup $l$ given by $l_k^+(\phi) = \{\mathbf{x} \in l | x_k \geq \phi\}$ and $l_k^-(\phi) = \{\mathbf{x} \in l | x_k < \phi\}$, where $\phi \in (x_k^{l,\text{min}}, x_k^{l,\text{max}})$ is the threshold for splitting, $x_k$ is the $k$-th covariate element, and $x_k^{l,\text{min}}$ and $x_k^{l,\text{max}}$ are the minimum and maximum values of the $k$-th covariate within the subgroup $l$, respectively.

We then apply SCR to each of these subsets. Specifically, we split the samples corresponding to $l_k^+(\phi)$ and $l_k^-(\phi)$ into training and validation sets: $\mathcal{D}_{l_k^+(\phi)} = \mathcal{I}_1^{l_k^+(\phi)} \cup \mathcal{I}_2^{l_k^+(\phi)}$ and $\mathcal{D}_{l_k^-(\phi)} = \mathcal{I}_1^{l_k^-(\phi)} \cup \mathcal{I}_2^{l_k^-(\phi)}$, where the split subsets are $\mathcal{I}_1^{l_k^+(\phi)} = \{(\mathbf{x}_i, y_i) \in \mathcal{I}_1^l : \mathbf{x}_i \in l_k^+(\phi)\}$ and $\mathcal{I}_2^{l_k^+(\phi)} = \{(\mathbf{x}_i, y_i) \in \mathcal{I}_2^l : \mathbf{x}_i \in l_k^+(\phi)\}$ (same for $l_k^-(\phi)$). To compute residuals, we do not train new estimators for $l_k^+(\phi)$ and $l_k^-(\phi)$; instead we use the previously trained estimator $\hat{\mu}_\mathcal{X}$; this provides consistency of estimators across groups and within groups. (It also avoids the enormous computational burden of training new estimators for all the possible splits.) Using the residuals, we can construct the confidence intervals $\hat{C}_{l_k^+(\phi)}(\mathbf{x})$ and $\hat{C}_{l_k^-(\phi)}(\mathbf{x})$ and the associated quantities in the objective function, $\hat{W}_{l_k^+(\phi)}, \hat{W}_{l_k^-(\phi)}, \hat{S}_{l_k^+(\phi)}$ and $\hat{S}_{l_k^-(\phi)}$. Then we find the optimal covariate $k_l^*$ and threshold $\phi_l^*$ for splitting subgroup $l$ as

$$(k_l^*, \phi_l^*) = \underset{(k,\phi)}{\operatorname{argmin}} \; \lambda \left( \hat{W}_{l_k^+(\phi)} + \hat{W}_{l_k^-(\phi)} \right) + (1-\lambda) \left( \hat{S}_{l_k^+(\phi)} + \hat{S}_{l_k^-(\phi)} \right).$$

For $(k_l^*, \phi_l^*)$, we compute $\hat{W}_{l\pm}^* = \hat{W}_{l_{k^*}^+(\phi^*)} + \hat{W}_{l_{k^*}^-(\phi^*)}$ and $\hat{S}_{l\pm}^* = \hat{S}_{l_{k^*}^+(\phi^*)} + \hat{S}_{l_{k^*}^-(\phi^*)}$. To improve the objective in (3), we split the subgroup $l$ into $l_{k^*}^+(\phi^*)$ and $l_{k^*}^-(\phi^*)$ only if the reduction in the impurity of the confident homogeneity is sufficiently large:

$$(1-\gamma) \left[ \lambda \hat{W}_l + (1-\lambda)\hat{S}_l \right] \geq \lambda \hat{W}_{l\pm}^* + (1-\lambda)\hat{S}_{l\pm}^* \tag{4}$$

Here, $\gamma \in [0,1)$ is a hyperparameter for regularization. We refer to (4) as the *confident criterion*. With an appropriate choice of $\gamma$, this criterion prevents overfitting, prevents the number of subgroups from becoming too large and prevents the size of each subgroup from becoming too small. Confident homogeneity does not necessarily improve as the group size shrinks because smaller groups lead to greater uncertainty. This alleviates the issue of generalization to unseen data in HTE analysis [8, 10].

After the splitting decision, we remove $l$ from $\Pi_c$; if we have split $l$, we remove $l$ from $\Pi$ and add the two split sets to both $\Pi$ and $\Pi_c$. We continue recursively until $\Pi_c$ is empty, at which point no further splitting is productive. When the procedure stops, we will have obtained an estimator $\mu_\mathcal{X}$ and a partition $\Pi$ and for each $l \in \Pi$ we will have corresponding confidence intervals $\hat{C}_l(\mathbf{x}) = \left[ \hat{\mu}_\mathcal{X}(\mathbf{x}) - \hat{Q}_{1-\alpha}^l, \hat{\mu}_\mathcal{X}(\mathbf{x}) + \hat{Q}_{1-\alpha}^l \right]$, where $\hat{Q}_{1-\alpha}^l$ is the $(1-\alpha)(1+1/|\mathcal{I}_2^l|)$-th quantile of the set of the residuals on the validation set $\mathcal{I}_2^l$ using $\hat{\mu}_\mathcal{X}$, $\{|y_i - \hat{\mu}_\mathcal{X}(\mathbf{x}_i)|\}_{i \in \mathcal{I}_2^l}$. The following theorem guarantees that the R2P partition Algorithm 1 provides a valid confidence interval $\hat{C}_l$ for each subgroup $l \in \Pi$; this is exactly what a user would want in subgroup analysis. The proof is provided in the supplementary material.

**Theorem 1** *Given a prescribed miscoverage rate $\alpha$, the created partition $\Pi$, estimator $\hat{\mu}_\mathcal{X}$, and confidence interval function $\hat{C}_l(\cdot)$ have the following property: for each $l \in \Pi$ and for new samples $(\mathbf{x}, y)$ drawn from the truncated distribution $\mathcal{P}_{X,Y}^l$, we have $\mathbb{P}[y \in \hat{C}_l(\mathbf{x})] \geq 1 - \alpha$.*

---

**Algorithm 1** Robust Recursive Partitioning

1: **Input**: Samples $\mathcal{D} = \{(\mathbf{x}_i, y_i)\}_{i=1}^n$, miscoverage rate $\alpha \in (0,1)$, $\Pi = \{\mathcal{X}\}$
2: **Initialize**: $\Pi_c = \Pi$, split $\mathcal{D}$ into $\mathcal{I}_1^\mathcal{X}$ and $\mathcal{I}_2^\mathcal{X}$, train $\hat{\mu}_\mathcal{X}$, compute its confidence interval $\hat{C}_\mathcal{X}$ using the split subsets, and obtain $\hat{W}_\mathcal{X}$ and $\hat{S}_\mathcal{X}$ using $\mathcal{I}_2^\mathcal{X}$
3: **for** $l \in \Pi_c$ **do**
4:     Obtain $\hat{W}_{l\pm}^*, \hat{S}_{l\pm}^*, i^*$, and $\phi^*$
5:     **if** $(1-\gamma)\left[ \lambda \hat{W}_l + (1-\lambda)\hat{S}_l \right] \geq \lambda \hat{W}_{l\pm}^* + (1-\lambda)\hat{S}_{l\pm}^*$ **then**         ▷ *Confident* criterion
6:         Partition $l$ into $l^+(i^*, \phi^*)$ and $l^-(i^*, \phi^*)$
7:         $\Pi \leftarrow \Pi \setminus \{l\}$
8:         $\Pi \leftarrow \Pi \cup \{l^+(i^*, \phi^*), l^-(i^*, \phi^*)\}$ and $\Pi_c \leftarrow \Pi_c \cup \{l^+(i^*, \phi^*), l^-(i^*, \phi^*)\}$
9:     **end if**
10:    $\Pi_c \leftarrow \Pi_c \setminus \{l\}$
11: **end for**
12: **Output**: $\Pi, \hat{\mu}_\mathcal{X}$, and $\hat{C}_l$ for all $l \in \Pi$

---

# 3 Robust Recursive Partitioning for Heterogeneous Treatment Effects

In this section, we extend the R2P method to the setting of HTE estimation, resulting in the *R2P-HTE* design as detailed below.

**Heterogeneous Treatment Effect Model**    We consider a setup with $n$ units (samples), For the unit $i \in \{1, 2, ..., n\}$, there exists a pair of potential outcomes, $Y_i(1)$ and $Y_i(0)$ that are independently drawn from an unknown distribution, where 0 and 1 represent whether the unit is treated or not, respectively. We define the treatment indicator as $t_i \in \{0, 1\}$, where $t_i = 1$ and 0 indicate the unit $i$ is treated and untreated, respectively. The outcome for unit $i$ is realized as the potential outcome corresponding to its treatment indicator $y_i = Y_i(t_i)$. A dataset is given as $\mathcal{D}_{\text{HTE}} = \{(\mathbf{x}_i, t_i, y_i)\}_{i=1}^n$. The ITE for a given $\mathbf{x}$ is defined as $\tau(\mathbf{x}) = \mathbb{E}[Y(1) - Y(0)|X = \mathbf{x}]$. Since the ITE is defined as the expected difference between the two potential outcomes $Y(1)$ and $Y(0)$, its estimator $\hat{\tau}(\mathbf{x})$ is given as the contrast between two regression models: $\hat{\mu}^0(\mathbf{x})$ for the conditional non-treated outcome $\mathbb{E}[Y(0)|X = \mathbf{x}]$, and $\hat{\mu}^1(\mathbf{x})$ for the conditional treated outcome $\mathbb{E}[Y(1)|X = \mathbf{x}]$.

**R2P-HTE**    We adapt R2P to HTE estimation by constructing the quantities $W_l$ and $S_l$ in (3) based on the ITE estimator $\hat{\tau}(\mathbf{x})$. To construct an ITE estimator, many popular machine learning models have been considered in the literature [8, 13–17]. R2P-HTE can use one of these models to parameterize the outcome models $\hat{\mu}^0(\mathbf{x})$ and $\hat{\mu}^1(\mathbf{x})$. We set the target coverage rate of $\hat{\mu}^0(\mathbf{x})$ and $\hat{\mu}^1(\mathbf{x})$ as $\sqrt{1-\alpha}$. As in the previous section, we can construct a confidence interval for each estimator by using the split conformal regression. Let us denote the $\sqrt{1-\alpha}$ confidence intervals for $Y(1)$ and $Y(0)$ by $\hat{C}^1(\mathbf{x}) = \left[\hat{\mu}^1(\mathbf{x}) - \hat{Q}^1_{\sqrt{1-\alpha}}, \hat{\mu}^1(\mathbf{x}) + \hat{Q}^1_{\sqrt{1-\alpha}}\right]$ and $\hat{C}^0(\mathbf{x}) = \left[\hat{\mu}^0(\mathbf{x}) - \hat{Q}^0_{\sqrt{1-\alpha}}, \hat{\mu}^0(\mathbf{x}) + \hat{Q}^0_{\sqrt{1-\alpha}}\right]$, respectively. We set the confidence interval for $\tau(\mathbf{x})$ to be

$$\hat{C}^\tau(\mathbf{x}) = \left[\hat{\mu}^1(\mathbf{x}) - \hat{\mu}^0(\mathbf{x}) - \hat{Q}^1_{\sqrt{1-\alpha}} - \hat{Q}^0_{\sqrt{1-\alpha}}, \hat{\mu}^1(\mathbf{x}) - \hat{\mu}^0(\mathbf{x}) + \hat{Q}^1_{\sqrt{1-\alpha}} + \hat{Q}^0_{\sqrt{1-\alpha}}\right].$$

This confidence interval ensures the coverage rate $1 - \alpha$ for the estimated ITE $\hat{\tau}(\mathbf{x}) = \hat{\mu}^1(\mathbf{x}) - \hat{\mu}^0(\mathbf{x})$, because its upper endpoint is given as the difference between the upper endpoint of $\hat{C}^1$ and the lower endpoint of $\hat{C}^0$, and its lower endpoint is given as the difference between the lower endpoint of $\hat{C}^1$ and the upper endpoint of $\hat{C}^0$. If the coverage rates for $\hat{C}^1$ and $\hat{C}^0$ are $\sqrt{1-\alpha}$, the coverage rate for $\hat{C}^\tau$ will be $1 - \alpha$.

From the ITE estimator $\hat{\tau}(\mathbf{x})$ and its confidence interval $\hat{C}^\tau_l(\mathbf{x})$ for each subgroup $l$, we can calculate $W_l$ and $S_l$ and adapt the R2P method to HTE estimation. The robust partitioning problem for HTE in (3) is solved by applying the R2P method in Algorithm 1, with two minor changes: 1) each sample in the HTE dataset is a triple $(\mathbf{x}_i, t_i, y_i)$ consisting of the covariate vector, the treatment indicator, and the observed outcome; 2) the outcome model $\hat{\mu}(\mathbf{x})$ in R2P is replaced by the ITE estimator $\hat{\tau}(\mathbf{x})$. As before, we show that this procedure achieves the specified coverage guarantee. The proof is provided in the supplementary material.

**Theorem 2** *Given a prescribed miscoverage rate $\alpha$, the created partition $\Pi$, estimator $\hat{\tau}_\mathcal{X}$, and confidence interval function $\hat{C}^\tau_l(\cdot)$ have the following property: for each $l \in \Pi$ and for new samples $(\mathbf{x}, \tau)$ drawn from the truncated distribution of the subgroup $l$, we have $\mathbb{P}[\tau \in \hat{C}^\tau_l(\mathbf{x})] \geq 1 - \alpha$.*

**Well-identified subgroups and false discovery**    Theorem 2 guarantees that confidence intervals achieve the required finite sample coverage for the ITE estimates in each subgroup, regardless of how (in)accurate the underlying ITE estimator $\hat{\tau}_\mathcal{X}$ is. If the confidence intervals exhibit large overlap across the constructed subgroups, we can conclude that the constructed subgroups are not well-identified. Conversely, if the confidence intervals have little or no overlap across subgroups, we conclude that the subgroups are well-identified. Given the theoretical guarantee of the confidence intervals in R2P, the subgroups are robust against false discoveries if they are well-identified.

# 4 Related Work

Subgroup analysis methods with recursive partitioning have been widely studied based on regression trees (RT) [8–11]. In these methods, the subgroups (i.e., leaves in the tree structure) are constructed and the individualized outcome or treatment effects are estimated by the corresponding sample mean estimator to the leaf for given covariates. To overcome the limitations of the traditional trees to

represent the non-linearity such as interactions between treatment and covariates [20], a parametric model is integrated into regression trees for subgroup analysis [12]. However, such approach can be used only for the limited types of estimator models, which is particularly undesirable since advanced causal inference models based on deep neural networks or multi-task Gaussian processes have been studied which outperform the traditional estimators [14–17]. The global model interpretation method in [21] can analyze the subgroup structure of arbitrary models but it depends on local model interpreters and does not consider the treatment effects.

For recursive partitioning, various criteria have been proposed. In the traditional RT, the criterion based on the mean squared error between the sample mean estimations from the training samples and the test samples is used [11], and it is referred to as the adaptive criterion. Based on the adaptive criterion, an honest criterion is proposed in [8] by splitting the training samples into the training set and the estimation set to eliminate the bias of the adaptive criterion. In addition, a generalization cost is introduced to the adaptive or honest criterion in [10] to encourage generalization of the analysis. The interaction measure between the treatment and covariates is used as a partitioning criterion in [9] and the parameter instability of the parametric models is used in [12]. In [21], the contribution matrix of the samples from local model interpreters is used. Some of these criteria implicitly consider the confidence of the estimation by the variance, but most of them do not provide a valid confidence interval of the estimation for each subgroup. In [11], a conformal regression method based on regression trees that provides the confidence interval is proposed, but the adaptive criterion is used for partitioning without any consideration of the confidence interval.

## 5  Experiments

In this section, we evaluate R2P-HTE by comparing its performance with state-of-the-art HTE methods. Specifically, we compare R2P-HTE with four baselines: standard regression trees for causal effects (CT-A) [22], conformal regression trees for causal effects (CCT) [11], causal trees with honest criterion (CT-H) [8], and causal trees with generalization costs (CT-L) [10]. We implement CCT and CT-A by modifying the conformal regression tree and conventional regression tree methods for causal effects. Details of the baseline algorithms are provided in the supplementary material. For the ITE estimator of R2P-HTE, here abbreviate as R2P, we use the causal multi-task Gaussian process (CMGP) [14]. Because individual ground-truth treatment effects can never be observed in real data, we use two synthetic and two semi-synthetic datasets. The first synthetic dataset (Synthetic dataset A) is the simple one proposed in [8]. Because dataset A possesses little of the homogeneity within subgroups that is often found in the real world, we offer a second synthetic dataset B that possesses greater homogeneity within subgroups and greater heterogenity across subgroups and has many more features than A. B was inspired by the initial clinical trial of remdesivir [23] as a treatment for COVID-19 and uses the patient features listed in that trial, but not the data. In specific, it represents a discovery that remdesivir results in a faster time to clinical improvement for the patients with the shorter time from symptom onset to starting trial. Aside from inspiration and features, B is relevant to COVID-19 only in that the COVID-19 is known to be a disease which displays in very heterogeneous ways. The two semi-synthetic datasets are based on real world data; the first uses the Infant Health and Development Program (IHDP) dataset [24] and the second uses the Collaborative Perinatal Project (CPP) dataset [25]. Details of all these datasets are provided in the supplementary material. For each experiment, we conduct 50 simulations.

The "optimal" ground-truth of subgroups depends on multiple objectives, including homogeneity, heterogeneity, and the number of subgroups. In the literature, a commonly adopted metric is the variance, because greater heterogeneity across subgroups and homogeneity within each subgroup generally imply well-discriminated subgroups. We denote the set of test samples by $\mathcal{D}^{te}$ and the test samples that belong to subgroup $l$ as $\mathcal{D}_l^{te}$. We define the mean and variance of the treatment effects of the test samples in subgroup $l$ as $\mathrm{Mean}(\mathcal{D}_l^{te})$ and $\mathrm{Var}(\mathcal{D}_l^{te})$, respectively. We define *heterogeneity across subgroups* to be the variance of the mean of the treatment effects: $V^{\mathrm{across}} = \mathrm{Var}(\{\mathrm{Mean}(\mathcal{D}_l^{te})\}_{l=1}^{L})$, where $L$ is the number of subgroups; we define the *average in-subgroup variance* to be $V^{\mathrm{in}}(\mathcal{D}^{te}) = \frac{1}{L}\sum_{l=1}^{L} \mathrm{Var}(\mathcal{D}_l^{te})$. We also provide the average number of subgroups for better understanding of the results. We set the miscoverage rate to be $\alpha = 0.05$, so we demand a 95% ITE coverage rate. (We do not report actual coverage rates here because all methods achieve the target coverage rate, but they are reported in the supplementary material, along with other details.)

**Results**  Table 1 reports the performances of R2P and the baselines for all four datasets. Keep in mind that *larger* $V^{\mathrm{across}}$ means greater heterogeneity *across* subgroups, while *smaller* $V^{\mathrm{in}}$ means

Table 1: **Comparison of methods** For the measures $V^{\text{across}}$ and $V^{\text{in}}$, and for the widths of confidence intervals, we highlight the best results in bold.

| | Synthetic dataset A | | | | Synthetic dataset B | | | |
|---|---|---|---|---|---|---|---|---|
| | $V^{\text{across}}$ | $V^{\text{in}}$ | # SGs | CI width | $V^{\text{across}}$ | $V^{\text{in}}$ | # SGs | CI width |
| R2P | **0.22±.01** | **0.03±.001** | 4.9±.16 | **0.08±.003** | **2.39±.04** | **0.12±.01** | 5.0±.16 | **0.88±.06** |
| CCT | 0.18±.02 | 0.05±.01 | 4.4±.24 | 7.42±.48 | 1.97±.14 | 0.58±.15 | 5.0±.13 | 5.95±.59 |
| CT-A | 0.19±.02 | 0.04±.01 | 4.7±.21 | 3.96±.16 | 2.24±.06 | 0.30±.05 | 5.1±.15 | 2.77±.20 |
| CT-H | 0.12±.03 | 0.11±.02 | 3.1±.39 | 4.39±.22 | 2.01±.13 | 0.53±.13 | 4.5±.15 | 3.38±.32 |
| CT-L | 0.12±.02 | 0.10±.02 | 2.9±.35 | 5.22±.02 | 0.80±.26 | 1.77±.27 | 3.1±.28 | 6.92±.53 |

| | IHDP dataset | | | | CPP dataset | | | |
|---|---|---|---|---|---|---|---|---|
| | $V^{\text{across}}$ | $V^{\text{in}}$ | # SGs | CI width | $V^{\text{across}}$ | $V^{\text{in}}$ | # SGs | CI width |
| R2P | **0.46±.04** | **0.38±.03** | 4.1±.12 | **1.27±.22** | **0.06±.02** | **0.10±.01** | 5.7±.30 | **1.11±.13** |
| CCT | 0.30±.04 | 0.43±.05 | 4.3±.13 | 5.70±.23 | 0.03±.02 | 0.12±.01 | 6.4±.20 | 3.60±.12 |
| CT-A | 0.31±.04 | 0.57±.05 | 4.1±.08 | 3.71±.08 | 0.03±.01 | 0.12±.01 | 6.6±.18 | 2.45±.06 |
| CT-H | 0.28±.05 | 0.56±.05 | 3.8±.14 | 3.76±.14 | 0.01±.00 | 0.14±.01 | 5.2±.23 | 2.67±.06 |
| CT-L | 0.27±.06 | 0.64±.05 | 2.8±.23 | 4.75±.15 | 0.01±.01 | 0.14±.01 | 2.9±.29 | 3.23±.07 |

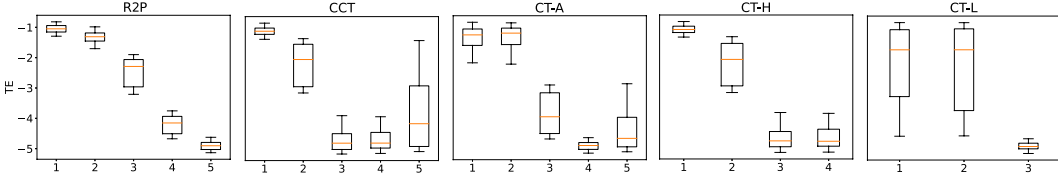

Figure 3: Treatment effects for subgroups identified by each algorithm when applied to Synthetic dataset B. Each box represents the range between the 25th and 75th percentiles of the treatment effects on the test samples; each whisker represents the range between the 5th and 95th percentiles.

greater homogeneity *within* subgroups. As the table shows, R2P displays by far the best performance on all four datasets: the greatest heterogeneity across subgroups, the greatest homogeneity within subgroups, and the narrowest confidence intervals. It accomplishes all this while identifying comparable numbers of subgroups. We conclude that R2P identifies subgroups more effectively than any of the other methods, and allows fewer false discoveries (as was illustrated in the Introduction). The performance of R2P reflects one of its strengths: the ability to use *any* method of estimating ITE.

The effectiveness of R2P can also be seen in Fig. 3, which provides, for R2P and each of the four baseline algorithms, boxplots of the distribution of the treatment effects for each identified subgroup for Synthetic dataset B. We see that that R2P identifies subgroups reliably: different subgroups have very different average treatment effects and their distributions are non-overlapping or well-discriminated. All the other methods are unreliable: false discovery occurs for all four baseline methods, and occurs frequently for three of the four.

**Gain from subgroup analysis** To indicate the gain from subgroup analysis obtained by R2P, and hence to indicate the effectiveness of recursive partitioning, we compare $V^{\text{in}}$, the homogeneity within subgroups obtained by R2P in Table 1, against the homogeneity within the

Table 2: Normalized $V^{\text{in}}$ of R2P

| Synth A | Synth B | IHDP | CPP |
|---|---|---|---|
| 0.110±.005 | 0.046±.003 | 0.459±.033 | 0.691±.076 |

entire population, $V^{\text{pop}}$. We divide $V^{\text{in}}$ by $V^{\text{pop}}$ to obtain the normalized $V^{\text{in}}$ in Table 2. Subgroup analysis with R2P reduces the average in-subgroup variance by 89% and more than 95% on Synthetic dataset A and B, respectively. For the semi-synthetic datasets, it reduces the average in-subgroup variance by more than 50% and 30% on the IHDP and CPP datasets, respectively. (Keep in mind that R2P constructs subgroups in a way that produces *both* strong heterogeneity across subgroups and strong homogeneity within subgroups.)

# 6 Conclusion

In this paper, we have studied robust HTE analysis based on recursive partitioning. The algorithm proposed, R2P-HTE, recursively partitions the entire population by taking into account both heterogeneity across subgroups and homogeneity within subgroups, using the novel criterion of confident homogeneity that is based on the quantification of uncertainty of the ITE estimation. R2P-HTE robustly constructs subgroups and also provides confidence guarantees for each subgroup. Experiments for synthetic and semi-synthetic datasets (the latter based on real data) demonstrate that R2P-HTE outperforms state-of-the-art baseline algorithms in every dimension: greater heterogeneity across subgroups, greater homogeneity within subgroups, and narrower confidence intervals. One of the strengths of R2P-HTE is that it can employ *any* method for interpretable ITE estimation, including improved methods that will undoubtedly be developed in the future.

## Broader Impact

The understanding of treatment effects plays an important role in many areas, and especially in medicine and public policy. In both areas, it is often the case that the same treatment has different effects on different groups; hence subgroup analysis is called for. In medicine, subgroup analysis may make it possible to identify groups of patients (defined by covariates such as age, body mass index, blood pressure, etc.) suffering from a particular disease for whom a particular drug is effective and safe and other groups for whom the same drug is ineffective and unsafe. Similarly, subgroup analysis may make it possible to identify groups of patients for whom one course of treatment (e.g. a particular mode of radiotherapy or chemotherapy) is preferable (more likely to be successful with fewer side effects) to another. In public policy, subgroup analysis may make it possible to identify groups of people or geographic regions for which particular interventions (e.g., providing mosquito nets to combat malaria) are likely to be successful or unsuccessful. The method for subgroup analysis that is developed in this paper, R2P, is an enormous improvement over state-of-the-art methods and therefore has the potential to make enormous and widespread impact. Moreover, because R2P can make use of improvements in the underlying estimation methods, this impact may grow over time.

## Acknowledgments

This work was supported by GlaxoSmithKline (GSK), the US Office of Naval Research (ONR), and the National Science Foundation (NSF) 1722516. CS acknowledges the funding support from Kneron, Inc. We thank all reviewers for their comments and suggestions.

## Footnotes

[2]The code of R2P is available at: `https://bitbucket.org/mvdschaar/mlforhealthlabpub`.

[3]Recall that assuming exchangeability is weaker than assuming iid.

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
