[Supplementary Material]

# Supplementary Material for Robust Recursive Partitioning for Heterogeneous Treatment Effects with Uncertainty Quantification

## A    Preliminaries of Conformal Prediction

Here, we provide a basic idea of conformal prediction to help understanding. To this end, we introduce the following example in [1]. Let $z_1, ..., z_n, z$ be samples of a scalar random variable exchangeably drawn from a distribution $P_Z$ and $z_{(1)}, ..., z_{(n)}$ denote the order statistics of $z_1, ..., z_n$. We denote the $1 - \alpha$-th quantile of $z_{(1)}, ..., z_{(n)}$ as

$$\hat{Q}_{1-\alpha} = \begin{cases} z_{(\lceil (n+1)(1-\alpha) \rceil)}, & \text{if } \lceil (n+1)(1-\alpha) \rceil \leq n \\ \infty, & \text{otherwise} \end{cases}$$

The rank of $z$ among $z_i, ..., z$ is uniformly distributed over the set $\{1, ..., n+1\}$ under the exchangeability assumption that the joint distribution of $z_1, ..., z_n, z$ is invariant of the sampling order $z_i, ..., z$. Thus, for a given miscoverage level $\alpha \in [0, 1]$, we have $\mathbb{P}[z \leq \hat{Q}_{1-\alpha}] \leq 1 - \alpha$ by summing the uniform distribution up to $\hat{Q}_{1-\alpha}$.

This idea can be used in a regression problem with covariates $\mathbf{x} \in \mathcal{X}$, where $d$ is the dimension of the covariate vector, and outcomes $y \in \mathcal{Y}$. Specifically, with the regressor $\hat{\mu}$, the confidence interval for $y$ is given as

$$\hat{C}(\mathbf{x}) = \left[ \hat{\mu}(\mathbf{x}) - G_{1-\alpha}^{-1}, \hat{\mu}(\mathbf{x}) + G_{1-\alpha}^{-1} \right],$$

where $G$ is the empirical distribution of the fitted residuals on the training samples (i.e., $|y_i - \hat{\mu}(\mathbf{x}_i)|$, $i = 1, ..., n$) and $G_{1-\alpha}^{-1}$ is the $1 - \alpha$-th quantile of $G$. However, this method may undercover $y$ since the residuals on the training samples typically smaller than those on the test samples due to overfitting. To avoid this, Split Conformal Regression (SCR) is introduced which separates the samples for training and computing the residuals. In SCP, the training samples is split into two equal-size subsets $\mathcal{I}_1$ and $\mathcal{I}_2$, and one subset $\mathcal{I}_1$ is used to fit the regressor $\hat{\mu}_{\mathcal{I}_1}$ and another one $\mathcal{I}_2$ is used to compute the residuals for $\hat{\mu}_{\mathcal{I}_1}$, $R_{\mathcal{I}_2} = \{ |y_i - \hat{\mu}_{\mathcal{I}_1}(\mathbf{x}_i)| : (\mathbf{x}_i, y_i) \in \mathcal{I}_2 \}$. Based on the regressor and residuals, the confidence interval for $y$ with the regressor $\hat{\mu}_{\mathcal{I}_1}$ is given as

$$\hat{C}_{SCR} = \left[ \hat{\mu}_{\mathcal{I}_1}(\mathbf{x}) - \hat{Q}_{1-\alpha}^{\mathcal{I}_2}, \hat{\mu}_{\mathcal{I}_1}(\mathbf{x}) + \hat{Q}_{1-\alpha}^{\mathcal{I}_2} \right],$$

where $\hat{Q}_{1-\alpha}^{\mathcal{I}_2}$ is defined to be $(1 - \alpha)(1 + 1/|\mathcal{I}_2|)$-th quantile of $R_{\mathcal{I}_2}$ (i.e., $\lceil (|\mathcal{I}_2| + 1)(1 - \alpha) \rceil$-the smallest residual on $\mathcal{I}_2$). Under the only one assumption of the exchangeability of the training samples $\{(\mathbf{x}_i, y_i)\}_{i=1}^n$ and the testing sample $(\mathbf{x}, y)$, it satisfies the following theorem.

**Theorem A.1** *[1] If the samples $\{(\mathbf{x}_i, y_i)\}_{i=1}^n$ are exchangeable, then for a new sample $(\mathbf{x}_{n+1}, y_{n+1})$ drawn from $\mathcal{P}_{X,Y}$,*

$$\mathbb{P}[y \in \hat{C}_{SCR}(\mathbf{x})] \geq 1 - \alpha.$$

## B    Proof of Theorem 1

For subgroup $l$, let $\mathcal{P}_{X,Y}^l$ be the distribution on $l \times \mathcal{Y}$, which is the conditional distribution $\mathcal{P}_{X,Y|X \in l}$. According to Algorithm 1, we have the samples of subgroup $l$ from the entire samples consisting of two disjoint subsets as $\mathcal{D}_l = \mathcal{I}_1^l \cup \mathcal{I}_2^l$, where $\mathcal{I}_1^l$ is the samples that are used for training the estimator $\hat{\mu}_{\mathcal{X}}$ and $\mathcal{I}_2^l$ is the samples that are used for constructing the confidence interval $\hat{C}_l$. The samples in $\mathcal{D}_l$ are exchangeable since they are i.i.d. Thus, for a new sample $(\mathbf{x}_{n+1}, y_{n+1})$ drawn from $\mathcal{P}_{X,Y}^l$, we have $\mathbb{P}[y_{n+1} \in \hat{C}^l] \leq 1 - \alpha$ from Theorem A.1.

# C  Proof of Theorem 2

We have the samples of subgroup $l$ from the entire samples consisting of two disjoint subsets as $\mathcal{D}_{\text{HTE},l} = \mathcal{I}_1^l \cup \mathcal{I}_2^l$. For training the estimators $\hat{\mu}_{\mathcal{X}}^1$ and $\hat{\mu}_{\mathcal{X}}^0$, the samples in $\mathcal{I}_1^l$ are used. Also, the samples in $\mathcal{I}_2^l$ are used to construct the confidence intervals $\hat{C}_l^1$ and $\hat{C}_l^0$ with the miscoverage rate $\sqrt{1-\alpha}$ according to the treatment indicator of the samples. Since the samples are i.i.d., for a new sample with each potential outcome, the corresponding confidence interval satisfies the miscoverage rate $\sqrt{1-\alpha}$ from Theorem A.1 as

$$\mathbb{P}[Y(1) \in \hat{C}_l^1(\mathbf{x})] \geq \sqrt{1-\alpha} \text{ and } \mathbb{P}[Y(0) \in \hat{C}_l^0(\mathbf{x})] \geq \sqrt{1-\alpha}.$$

For the definitions of the ITE estimation $\hat{\tau}_l(\mathbf{x})$ and $\hat{C}_l^\tau(\mathbf{x})$, the events that the potential outcomes and ITE estimation belong to their corresponding confidence intervals satisfy the following relation:

$$\{Y(1) \in \hat{C}_l^1(\mathbf{x})\} \cap \{Y(0) \in \hat{C}_l^0(\mathbf{x})\} \subset \{\tau \in \hat{C}_l^\tau(\mathbf{x})\},$$

which implies that

$$\mathbb{P}[\tau \in \hat{C}_l^\tau(\mathbf{x})] \geq \mathbb{P}[Y(1) \in \hat{C}_l^1(\mathbf{x})] \times \mathbb{P}[Y(0) \in \hat{C}_l^0(\mathbf{x})] \geq 1-\alpha.$$

# D  Related Works in Subgroup Analysis with Recursive Partitioning

Subgroup analysis methods with recursive partitioning have been widely studied based on regression trees (RT) [2–5]. In these methods, the subgroups (i.e., leaves in the tree structure) are constructed; the treatment effects are estimated by the corresponding sample mean estimator on the leaf of the given covariates. To represent the non-linearity such as interactions between treatment and covariates [6], a parametric model is integrated into regression trees for subgroup analysis [7]. However, such approach can be used only for the limited types of models, which is not particularly satisfying given the fact that advanced causal inference models based on deep neural networks or multi-task Gaussian processes have been studied which outperform the traditional estimators [8–11]. The global model interpretation method in [12] can analyze the subgroup structure of arbitrary models but it depends on local model interpreters and does not consider the treatment effects.

Table 1: Comparison of related works

| Method | Estimator | Partitioning Criterion | | | Treatment effect |
|---|---|---|---|---|---|
| | | Type | Homogeneity | CI widths | |
| [2] | Sample mean | Honest criterion | $\times$ | $\times$ | $\checkmark$ |
| [4] | Sample mean | Adaptive criterion with generalization cost | $\times$ | $\times$ | $\checkmark$ |
| [5] | Sample mean | Adaptive criterion | $\times$ | $\times$ | $\times$ |
| [3] | Sample mean | Interaction measure | $\times$ | $\times$ | $\checkmark$ |
| [7] | Parametric model | Parameter instability | $\times$ | $\times$ | $\checkmark$ |
| [12] | Arbitrary estimator | Feature contribution | $\times$ | $\times$ | $\times$ |
| Our work | Arbitrary estimator | Confident criterion | $\checkmark$ | $\checkmark$ | $\checkmark$ |

For recursive partitioning, various criteria have been proposed. In the traditional RT, the criterion based on the mean squared error of the estimated means on the training samples and the test samples is used [5], and it is referred to as the adaptive criterion. Basically, this adaptive criterion identifies subgroups with heterogeneous treatment effects by trying to maximize the heterogeneity across the identified subgroups. Based on the adaptive criterion, in [2], an honest criterion is proposed. In the criterion, the training samples are split into two subsets; One is used to build a tree structure and another one is used to estimate the treatment effects. By doing this, the honest criterion can eliminate the bias of the adaptive criterion. In [4], a generalization cost is introduced to encourage generalization of the analysis. It is defined by using another subset of the training samples and

adopted to the adaptive or honest criterion. The interaction measure between the treatment and covariates is used as a partitioning criterion in [3] and the parameter instability of the parametric models is used in [7]. In [12], the contribution matrix of the samples from local model interpreters is used for partitioning. These criteria focus on the heterogeneity across the subgroups, but the variant of the treatment effects within each subgroup (i.e., the homogeneity within each subgroup) is neglected. Besides, some of these criteria construct the confidence intervals using the estimated variances, but most of these intervals fail to achieve the coverage guarantee for each subgroup in finite samples. In [5], a conformal regression method for constructing confidence intervals using regression trees is proposed. The adaptive criterion they use for partitioning does not take into account the confidence interval. The confident criterion in R2P is different from these criteria by considering both heterogeneity and homogeneity of subgroups and constructing subgroups with confidence intervals.

# E  Description of Datasets

## E.1  Description of Synthetic Models

**Synthetic dataset A**  We first consider the synthetic treatment effect model proposed in [2]. It describes the potential outcome $y$ for given treatment $t \in \{0, 1\}$ as follows:

$$y_i(t) = \eta(\mathbf{x}_i) + \frac{1}{2}(2t - 1)\kappa(\mathbf{x}_i) + \epsilon_i,$$

where $\epsilon_i \sim \mathcal{N}(0, 0.01)$, $x_{i,k} \sim \mathcal{N}(0, 1)$, and $\eta(\cdot)$ and $\kappa(\cdot)$ are the design functions. The response surface of outcome $y_i$'s is determined by the design functions. The functions $\eta(\cdot)$ and $\kappa(\cdot)$ are the mean outcome and treatment effect for some given covariates, respectively. In this synthetic model, we consider the following design with 2-dimensional covariates, $\eta(\mathbf{x}) = \frac{1}{2}x_1 + x_2$ and $\kappa(\mathbf{x}) = \frac{1}{2}x_1$. There is no redundant covariate which has no effect on the outcomes. In the experiments, we generate 300 samples for training and 1000 samples for testing.

**Synthetic dataset B**  Here, we introduce a synthetic model based on the initial clinical trial results of remdesivir to COVID-19 [13]. The result shows that remdesivir results in a faster time to clinical improvement for the patients with a shorter time from symptom onset to starting the trial. Since the clinical trial data is not public, we generate a synthetic model in which the treatment effects mainly depends on the time from symptom onset to trial based on the clinical trial setting and results in the paper [13]. We consider the following 10 baseline covariates: age $\sim \mathcal{N}(66, 4)$, white blood cell count ($\times 10^9$ per L) $\sim \mathcal{N}(66, 4)$, lymphocyte count ($\times 10^9$ per L) $\sim \mathcal{N}(0.8, 0.1)$, platelet count ($\times 10^9$ per L) $\sim \mathcal{N}(183, 20.4)$, serum creatinine (U/L) $\sim \mathcal{N}(68, 6.6)$, asparatate aminotransferase (U/L) $\sim \mathcal{N}(31, 5.1)$, alanine aminotransferase (U/L) $\sim \mathcal{N}(26, 5.1)$, lactate dehydrogenase (U/L) $\sim \mathcal{N}(339, 51)$, creatine kinase (U/L) $\sim \mathcal{N}(76, 21)$, and time from symptom onset to starting the trial (days) $\sim \mathrm{Unif}(4, 14)$. We approximate the distribution using the patient characteristics provided in the paper. To construct treatment/control responses, we first adopt the response surface in the IHDP dataset [14] for the covariates except for the time. We then use a logistic function on the time covariate to produce different effectiveness (i.e., the faster time to clinical improvement with the shorter time from symptom onset to the trial). Specifically, the control response is defined as $Y(0) \sim \mathcal{N}(X_{-0}\boldsymbol{\beta} + (1 + e^{-(x_0 - 9)})^{-1} + 5, 0.1)$, and the treated response is defined as $Y(1) \sim \mathcal{N}(X_{-0}\boldsymbol{\beta} + 5 \cdot (1 + e^{-(x_0 - 9)})^{-1}, 0.1)$, where $X_{-0}$ represents the matrix of the standardized (zero-mean and unit standard deviation) covariate values except for the time covariate $x_0$ and the coefficients in the vector $\boldsymbol{\beta}$ are randomly sampled among the values $(0, 0.1, 0.2, 0.3, 0.4)$ with the probability $(0.6, 0.1, 0.1, 0.1, 0.1)$, respectively. In this synthetic model, the response surface is consistent with the trial result in [13] such that the time to clinical improvement (i.e., the treatment effect) becomes faster as the shorter time from symptom onset to the trial.

## E.2  Description of Semi-Synthetic Datasets

We consider two semi-synthetic datasets for treatment effect estimation: the Infant Health and Development Program (IHDP) [14] and the Collaborative Perinatal Project (CPP) [15].

**IHDP dataset**  The Infant Health and Development Program (IHDP) is a randomized experiment intended to enhance the cognitive and health status of low-birth-weight, premature infants through

intensive high-quality child care and home visits from a trained provider. Based on the real experimental data about the impact of the IHDP on the subjects' IQ scores at the age of three, the semi-synthetic (simulated) dataset is developed and has been used to evaluate treatment effects estimation in [14, 8, 10, 16]. All outcomes (i.e., response surfaces) are simulated using the real covariates. The dataset consists of 747 subjects (608 untreated and 139 treated), and 25 input covariates for each subject. We generated the outcomes using the standard non-linear mean outcomes of "Response Surface B" setting provided in [14]. A noise $\epsilon \sim \mathcal{N}(0, 0.1)$ is added to each observed outcome. In the experiments, we use 80% samples for training and 20% samples for testing.

**CPP dataset**  In the 2016 Atlantic Causal Inference Conference competition (ACIC), a semi-synthetic dataset is developed based on the data from the Collaborative Perinatal Project (CPP) [15]. It comprises of multiple datasets that are generated by distinct data generating processes (causal graphs) and random seeds. Each dataset consists of 4802 observations with 58 covariates of which 3 are categorical, 5 are binary, 27 are count data, and the remaining 23 are continuous. The factual and counterfactual samples are drawn from a generative model and a noise $\epsilon \sim \mathcal{N}(0, 0.1)$ is added for each observed outcome. In the experiments, we use the dataset with index 1 provided in [15] and drop the rows whose $Y(1)$ or $Y(0)$ above the 99% quantile or below the 1% quantile to avoid the outliers. The dataset consists of 35% treated subjects and 65% untreated subjects. We randomly pick 500 samples for training and 300 samples for testing from the dataset.

## F  Description of Algorithms in Experiments

**Robust recursive partitioning for HTE (R2P-HTE)**  We implement R2P-HTE based on Section 4 of the main manuscript. For the ITE estimator, we use the causal multi-task Gaussian process (CMGP) in [8]. Using the outcome estimates from CMGP (i.e., $\hat{\mu}^1(\mathbf{x})$ and $\hat{\mu}^0(\mathbf{x})$), we construct the confidence interval for the ITE estimator $\hat{\tau}(\mathbf{x})$. Then, we can apply Algorithm 1 in the main manuscript. In the experiments, we set $\lambda = 0.5$ and $\gamma = 0.05$. (We use $\lambda = 0$ for the CPP dataset to avoid the excessive effect of the confidence intervals compared with the heterogeneity effect in the dataset.) We use $\alpha = 0.1$ and 0.5 split ratio for split conformal prediction. We set the minimum number of training samples in each leaf as 10. To avoid excessive conservativeness, we use the confidence interval of the miscoverage rate $\beta = 0.8$ for the expected deviation, $\hat{S}_l$, in the confident split criterion. We set the hyper-parameters manually as above considering a typical value (e.g., 0.95 is a typical value for significance tests in tree-based methods), but if needed, the hyper-parameter tuning can be done by a grid search method as in a typical recursive partitioning methods [4].

**Standard regression tree for causal effects (CT-A)**  Because a standard regression tree in [17] is not developed for estimating treatment effects, we implement the modified version of the standard regression tree for causal effects estimation in [2]. In this modified version, the regression tree recursively partitions according to a criterion based on the expectation of the mean squared error (MSE) of the treatment effects. In the literature, this criterion called an adaptive criterion. We refer to [2] for more details of the method. In the experiments, we set the minimum number of training samples in each leaf as 20 since CT-A does not need to split the data samples into two subsets for validation as in other methods. After building the tree-structure, we prune the tree according to the statistical significance gain at 0.05.

**Conformal regression tree for causal effects (CCT)**  We modify the conformal regression tree [5] for our experiments of treatment effect estimation. We implemented CCT by applying the split conformal prediction method to a standard causal tree (i.e., CT-A). The ITE confidence interval is constructed in the same way as R2P. In the experiments, we use 0.5 split ratio for the split conformal prediction method and set the minimum number of training samples in each leaf as 10. After building the tree-structure, we prune the tree according to the statistical significance gain at 0.05.

**Causal tree with honest criterion (CT-H)**  We implement a causal tree method proposed in [2]. The method modify the standard regression tree for causal effects in which an honest criterion is used instead of the adaptive criterion. It divides tree-building and treatment effect estimation into two steps. The samples are split into two subsets: training samples to build the tree and samples to estimate treatment effects. This two-step procedure makes the tree-building and the treatment effect estimation process independent, which can eliminate the bias in treatment effect estimation. We refer

to [2] for more details of the method, In the experiments, we use 0.5 split ratio for building the tree and estimating the effects. We set the minimum number of training samples in each leaf as 10. After building the tree-structure, we prune the tree according to the statistical significance gain at 0.05.

**Causal tree with generalization costs (CT-L)**   We implement a causal tree with a criterion considering generalization costs in [4]. This method is a modified version of the causal tree in [2]. It splits the data samples into the training and validation samples, and builds the tree using the training samples while penalizing based on generalization ability using the validation samples. For more details of the method, we refer to [4]. In the experiments, we use 0.5 split ratio for the training and validation. We set the minimum number of training samples in each leaf as 10. After building the tree-structure, we prune the tree according to the statistical significance gain at 0.05.

## G   Additional Experimental Results

### G.1   Average Overlap of Treatment Effects across Subgroups

As one metric for evaluating false discovery, we can use the overlap of treatment effects across subgroups in Fig. 3 of the main manuscript. The average overlap of treatment effects across subgroups indicates whether the subgroups are well-discriminated. Specifically, we define a treatment effect interval of each subgroup $l$ as $[a_l(p), b_l(q)]$, where $a_l(p)$ and $b_l(q)$ are $p$-th and $q$-th percentiles of the treatment effects in the subgroup $l$. We define the average overlap

Table 2: Average overlap of treatment effects across subgroups.

|       | Synth A | Synth B | IHDP | CPP |
|-------|---------|---------|------|-----|
| R2P   | **0.45±.06** | **0.14±.03** | **0.32±.04** | **0.23±.03** |
| CCT   | 1.35±.04 | 0.63±.15 | 0.81±.09 | 0.55±.04 |
| CT-A  | 1.13±.21 | 0.44±.09 | 0.59±.08 | 0.47±.05 |
| CT-H  | 0.60±.20 | 0.60±.16 | 0.76±.10 | 0.45±.05 |
| CT-L  | 0.87±.18 | 2.27±.55 | 0.46±.10 | 0.24±.04 |

of treatment effects across subgroups as the overlapped width of the treatment effect intervals between all the pairs of the subgroups. We provide the average overlaps of R2P and the baselines for all datasets with $p = 20$ and $q = 80$. The table shows that the average overlap width of R2P is significantly small than the one of the baselines, which implies that R2P performs best for discriminating the subgroups.

### G.2   Results with Maximum Depth for Partitioning

We provide the results of the maximum depth for partitioning in R2P to demonstrate the effectiveness of the confident criterion more clearly. We set the maximum depth of each method to be 2, which limits the maximum number of identified subgroups by 4. In most datasets, R2P has both the highest variance across subgroups and lowest in-subgroup variance. This implies that each partitioning in R2P is more effective to identify the subgroups than that in the other methods. In Synthetic dataset B, CT-A has the highest variance across subgroups, but its difference from that of R2P is marginal and the in-subgroup variance of CT-A is much higher than that of R2P.

Table 3: Results with maximum depth for partitioning.

|       | Synthetic dataset A | | | | | Synthetic dataset B | | | | |
|-------|---------|--------|-------|----------|----------|---------|--------|-------|----------|----------|
|       | $V^{\text{across}}$ | $V^{\text{in}}$ | # SGs | CI width | Cov. (%) | $V^{\text{across}}$ | $V^{\text{in}}$ | # SGs | CI width | Cov. (%) |
| R2P   | **0.27±.01** | **0.04±.001** | 4.0±.04 | **0.09±.002** | 99.06±.23 | 2.19±.04 | **0.14±.01** | 4.0±.00 | **0.98±.07** | 99.28±.16 |
| CCT   | 0.20±.02 | 0.07±.01 | 3.4±.22 | 8.28±.42 | 100.0±.00 | 2.10±.08 | 0.43±.09 | 3.9±.09 | 6.05±.46 | 99.99±.01 |
| CT-A  | 0.24±.02 | 0.06±.01 | 3.6±.15 | 4.29±.16 | 99.99±.02 | **2.23±.06** | 0.30±.06 | 3.9±.08 | 2.97±.20 | 98.68±.52 |
| CT-H  | 0.14±.03 | 0.11±.02 | 2.5±.27 | 4.71±.18 | 99.99±.01 | 2.13±.09 | 0.45±.09 | 3.8±.12 | 3.45±.25 | 98.66±.68 |
| CT-L  | 0.13±.03 | 0.12±.02 | 2.4±.25 | 5.49±.16 | 99.99±.01 | 0.82±.23 | 1.74±.25 | 2.8±.19 | 6.71±.53 | 99.70±.30 |

|       | IHDP dataset | | | | | CPP dataset | | | | |
|-------|---------|--------|-------|----------|----------|---------|--------|-------|----------|----------|
|       | $V^{\text{across}}$ | $V^{\text{in}}$ | # SGs | CI width | Cov. (%) | $V^{\text{across}}$ | $V^{\text{in}}$ | # SGs | CI width | Cov. (%) |
| R2P   | **0.52±.05** | **0.42±.05** | 3.5±.14 | **1.21±.13** | 97.24±.50 | **0.05±.02** | **0.12±.01** | 3.7±.13 | **1.22±.14** | 99.04±.23 |
| CCT   | 0.28±.05 | 0.62±.06 | 3.5±.14 | 6.11±.21 | 99.55±.14 | **0.05±.03** | 0.13±.01 | 3.4±.14 | 3.66±.13 | 99.50±.21 |
| CT-A  | 0.33±.04 | 0.58±.05 | 3.7±.13 | 3.64±.08 | 97.25±.43 | 0.02±.01 | 0.13±.01 | 3.5±.14 | 2.44±.05 | 96.17±.51 |
| CT-H  | 0.30±.05 | 0.60±.05 | 3.5±.14 | 3.72±.12 | 97.17±.43 | 0.01±.00 | 0.14±.01 | 3.4±.14 | 2.59±.06 | 97.31±.56 |
| CT-L  | 0.30±.06 | 0.67±.04 | 2.7±.18 | 4.72±.17 | 98.99±.23 | 0.02±.02 | 0.13±.01 | 2.7±.21 | 3.25±.07 | 99.55±.22 |

## G.3 Results of R2Ps with Different ITE Estimators

Here, we provide the results of R2Ps with different ITE estimators in Table 4. For this, we integrate R2P with the ITE estimators based on dragonnet (DN) [18], random forest (RF), and CT-H [2]. To evaluate the precision of the ITE estimation, we introduce a precision in estimation of heterogeneous effect (PEHE) defined as follows [14]:

$$\text{PEHE} = \frac{1}{n} \sum_{i=1}^{n} ((\hat{\mu}^1(X_i) - \hat{\mu}^0(X_i)) - \mathbb{E}[Y_i^{(1)} Y_i^{(1)} | X_i = \mathbf{x}])^2.$$

The lower PEHE implies a more accurate ITE estimation. In the table, we can see that the use of a better estimator allows R2P to construct better subgroups. This clearly shows that R2P can effectively exploit the better precision of the ITE estimation when constructing subgroups. Besides, R2P can seamlessly integrate any ITE estimators. Consequently, we can expect that R2P constructs better subgroups by using some improved ITE estimators in the future.

Table 4: Results of R2Ps with different ITE estimators.

| | Synthetic dataset A | | | | | Synthetic dataset B | | | | |
|---|---|---|---|---|---|---|---|---|---|---|
| | $V^{\text{across}}$ | $V^{\text{in}}$ | # SGs | CI width | $\sqrt{\text{PEHE}}$ | $V^{\text{across}}$ | $V^{\text{in}}$ | # SGs | CI width | $\sqrt{\text{PEHE}}$ |
| R2P | 0.22±.01 | 0.03±.001 | 4.9±.16 | 0.08±.003 | 0.01±.00 | 2.39±.04 | 0.12±.01 | 5.0±.16 | 0.88±.06 | 0.16±.01 |
| R2P-DN | 0.21±.02 | 0.05±.01 | 4.9±.27 | 0.71±.05 | 0.06±.00 | 2.32±.06 | 0.19±.03 | 5.1±.17 | 2.24±.09 | 0.41±.02 |
| R2P-RF | 0.18±.02 | 0.08±.01 | 4.9±.26 | 2.91±.12 | 0.33±.01 | 2.20±.08 | 0.33±.07 | 5.1±.15 | 3.10±.32 | 0.42±.04 |
| R2P-CT-H | 0.12±.02 | 0.07±.03 | 4.3±.49 | 8.87±.32 | 0.51±.02 | 1.05±.25 | 1.51±.25 | 4.6±.41 | 6.42±.51 | 0.92±.12 |

| | IHDP dataset | | | | | CPP dataset | | | | |
|---|---|---|---|---|---|---|---|---|---|---|
| | $V^{\text{across}}$ | $V^{\text{in}}$ | # SGs | CI width | $\sqrt{\text{PEHE}}$ | $V^{\text{across}}$ | $V^{\text{in}}$ | # SGs | CI width | $\sqrt{\text{PEHE}}$ |
| R2P | 0.46±.04 | 0.38±.03 | 4.1±.12 | 1.27±.22 | 0.22±.02 | 0.06±.02 | 0.10±.01 | 5.7±.30 | 1.11±.13 | 0.13±.01 |
| R2P-DN | 0.41±.03 | 0.44±.04 | 4.3±.13 | 1.92±.09 | 0.33±.01 | 0.06±.02 | 0.10±.01 | 6.0±.26 | 1.52±.07 | 0.19±.01 |
| R2P-RF | 0.32±.05 | 0.55±.05 | 4.0±.32 | 3.07±.13 | 0.39±.02 | 0.04±.02 | 0.12±.01 | 6.0±.27 | 1.80±.06 | 0.23±.01 |
| R2P-CT-H | 0.09±.04 | 0.75±.05 | 2.7±.50 | 6.56±.25 | 0.83±.03 | 0.00±.00 | 0.14±.00 | 1.0±.004 | 4.20±.10 | 0.41±.01 |

## G.4 Non-Interpretability of Grouping Using Quantiles of ITE Estimation

One naive way to construct subgroups is dividing the covariate space with respect to the quantiles of estimated ITEs. However, this approach fails to satisfy the essential requirement of subgroup analysis: interpretability. The estimates from a black-box ITE estimator are non-interpretable. Similarly, the subgroups defined based on the estimated quantiles give no explanation (in terms of input covariates) regarding why the samples are assigned to a particular subgroup. To demonstrate this problem clearly, in Fig. 1, we divide the covariate space of the IHDP dataset into four subgroups based on the intervals of quantiles of CMGP, $[0, 25)$, $[25, 50)$, $[50, 75)$ and $[75, 100]$. The colours indicate which subgroup each sample belongs to. We can see that the quantile fails to provide interpretability in terms of the input covariates. In contrast, R2P constructs easy-to-interpret subgroups based on tree-structure.

Figure 1: Subgroups in four different intervals of quantiles.

## G.5 Complete Results of Table 1 in the Manuscript

Below we provide the complete results of Table 1 in the main manuscript. We can see that R2P satisfies the target coverage for all datasets even with the narrower confidence interval widths.

## G.6 Impact of Hyper-Parameters

Here we show the impact of the hyper-parameters $\gamma$ and $\lambda$ of R2P to the performance. We provide the results of varying the hyper-parameters as $\gamma \in \{0.01, 0.02, 0.05, 0.1, 0.15, 0.2, 0.5\}$. and $\lambda \in \{0.1, 0.2, 0.3, 0.4, 0.5, 0.6, 0.7, 0.8, 0.9\}$. We use the same experiment setup in the main manuscript, and repeat the experiments 50 times for each hyper-parameter.

**Impact of hyper-parameter** $\gamma$  We show the impact of the hyper-parameter $\gamma \in [0, 1)$ of R2P in Fig. 2. The hyper-parameter $\gamma$ controls the regularization in R2P. From the figures, we can see that as $\gamma$ increases, the number of subgroups converges to one since a group is barely partitioned with the larger $\gamma$. This causes the degradation of performance in the following aspects: $V^{\text{across}}$ decreases, $V^{\text{in}}$ increases, and the confidence interval width increases generally. Thus, it seems that a smaller value is a better choice for $\gamma$. However, if $\gamma$ is too small, the subgroups (i.e., the partition) constructed by R2P becomes overfitted. This overfitting issue results in the loss of generalization ability on the unseen data, and a large number of subgroups due to the overfitting makes the subgroup analysis less informative. In addition, we can see that our method robustly satisfies the target coverage rate regardless of $\gamma$. Therefore, the hyper-parameter $\gamma$ should be appropriately chosen in practice.

Table 5: Complete results of Table 1 in the manuscript.

| | Synthetic dataset A | | | | | Synthetic dataset B | | | | |
|---|---|---|---|---|---|---|---|---|---|---|
| | $V^{\text{across}}$ | $V^{\text{in}}$ | # SGs | CI width | Cov. (%) | $V^{\text{across}}$ | $V^{\text{in}}$ | # SGs | CI width | Cov. (%) |
| R2P | **0.22±.01** | **0.03±.001** | 4.9±.16 | **0.08±.003** | 98.98±.24 | **2.39±.04** | **0.12±.01** | 5.0±.16 | **0.88±.06** | 98.86±.23 |
| CCT | 0.18±.02 | 0.05±.01 | 4.4±.24 | 7.42±.48 | 100.0±.00 | 1.97±.14 | 0.58±.15 | 5.0±.13 | 5.95±.59 | 99.86±.13 |
| CT-A | 0.19±.02 | 0.04±.01 | 4.7±.21 | 3.96±.16 | 99.99±.02 | 2.24±.06 | 0.30±.05 | 5.1±.15 | 2.77±.20 | 97.73±.76 |
| CT-H | 0.12±.03 | 0.11±.02 | 3.1±.39 | 4.39±.22 | 99.98±.02 | 2.07±.13 | 0.53±.13 | 4.5±.15 | 3.38±.32 | 98.20±.73 |
| CT-L | 0.12±.02 | 0.10±.02 | 2.9±.35 | 5.22±.02 | 99.97±.06 | 0.80±.26 | 1.77±.27 | 3.1±.28 | 6.92±.53 | 99.44±.47 |

| | IHDP dataset | | | | | CPP dataset | | | | |
|---|---|---|---|---|---|---|---|---|---|---|
| | $V^{\text{across}}$ | $V^{\text{in}}$ | # SGs | CI width | Cov. (%) | $V^{\text{across}}$ | $V^{\text{in}}$ | # SGs | CI width | Cov. (%) |
| R2P | **0.46±.04** | **0.38±.03** | 4.1±.12 | **1.27±.22** | 97.93±.39 | **0.06±.02** | **0.10±.01** | 5.7±.30 | **1.11±.13** | 98.52±.34 |
| CCT | 0.30±.04 | 0.53±.05 | 4.3±.13 | 5.70±.23 | 99.59±.12 | 0.03±.02 | 0.12±.01 | 6.4±.20 | 3.60±.12 | 99.54±.23 |
| CT-A | 0.31±.04 | 0.57±.05 | 4.1±.08 | 3.71±.08 | 97.41±.42 | 0.03±.01 | 0.12±.01 | 6.6±.18 | 2.45±.06 | 96.60±.50 |
| CT-H | 0.28±.05 | 0.56±.05 | 3.8±.14 | 3.76±.14 | 97.76±.40 | 0.01±.00 | 0.14±.01 | 5.2±.23 | 2.67±.06 | 98.01±.40 |
| CT-L | 0.27±.06 | 0.64±.05 | 2.8±.23 | 4.75±.15 | 98.97±.30 | 0.01±.01 | 0.14±.01 | 2.9±.29 | 3.23±.07 | 99.49±.23 |

(a) Results with Synthetic dataset A.

(b) Results with Synthetic dataset B.

(c) Results with IHDP dataset.

(d) Results with CPP dataset.

Figure 2: Results on varying $\gamma$. (the red dotted line corresponds to the target coverage rate).

**Impact of hyper-parameter** $\lambda$  We show the impact of the hyper-parameter $\lambda$ of R2P in Fig. 3. The hyper-parameter $\lambda \in [0, 1]$ controls the weight between the homogeneity within each subgroup and the confidence interval discrimination in the confident criterion. With smaller $\lambda$, the homogeneity

within each subgroup is more emphasized in the criterion. So in that case, from the figures, we can see that R2P finds the larger number of subgroups, which results in the higher $V^{\mathrm{across}}$ and the lower $V^{\mathrm{in}}$. On the other hand, with larger $\lambda$, the confidence interval discrimination is weighted higher in the criterion, and thus, the confidence interval width decreases generally. We can see that our method robustly satisfies the target coverage rate regardless of $\lambda$ in most cases. Overall, $\lambda$ should be appropriately chosen considering the variance over the entire population.

(a) Results with Synthetic dataset A.

(b) Results with Synthetic dataset B.

(c) Results with IHDP dataset.

Figure 3: Results on varying $\lambda$. (the red dotted line corresponds to the target coverage rate).