[Reviews · NeurIPS 2020]

Review 1

Summary and Contributions: Paper provides a subgrouping algorithm that estimates confidence intervals on mean outcome and then uses the interval bounds to produce improved subgroups. The idea is that confidence intervals will capture uncertainty in effect estimates which can then provide a way to control heterogeneity within a subgroup.

Strengths: The paper provides a new method for using confidence interval estimates to improve subgrouping. The idea is intuitive and seems to work well in the experiments. The theoretical claims are reasonable but could be better. The experiments are wide which adds to the merit of the method and the paper.

Weaknesses: The paper needs to do a much better job of what happens when the underlying mean outcome estimate is bad. For example, what if I predict random normal noise as mean outcome for each sample in the dataset? How does this affect the estimated subgroups? How small must the error be in ITE so that that subgroups are meaningful? Such error bounds would significantly improve the usability of R2P. Finally, using ITE estimates based on separate mean outcome estimates induces error due to covariate imbalance. How does this affect R2P-HTE?

Correctness: The claims are correct.

Clarity: The paper is well written.

Relation to Prior Work: The related work should be explained better. As in why is not relying on the ITE estimation algorithm better? Where does this help?

Reproducibility: Yes

Additional Feedback: Update after rebuttal: Thanks for clarifying the claims. I'm updating my score accordingly.


Review 2

Summary and Contributions: This paper presents an outlier detection approach based on the self-supervised training. The proposed approach composes from investigating the pretrained features and self-supervised learning. The paper develops a novel method for subgroup analysis, which is both more reliable and more informative than previous methods via promising confidence estimates. Experiments demonstrate its effectiveness compared with various baselines.

Strengths: The problem is meaningful and interesting. The structure of the paper is explicit, which is easy to follow and review. The methodology part is clearly presented and solid. Experimental results validate the effectiveness of the proposed model on both synthetic dataset and real-world dataset, especially on COVID 19 applications. Supplementary materials validates the effectiveness of the proposed methods with concrete experimental design and codes brings the reproducibility.

Weaknesses: The methodology part lacks some deeper understanding and innovations. How does quantifying the uncertainty benefit the ITE process, how did the proposed method specifically tailored to the given task. More relevant analysis is expected. The analysis of experimental results is not enough. Uncertainty usually brings crucial information to the target task. The proposed method quantifies the uncertainty in the methodology part, yet more deeper insights and analysis are expected in the experiment part.

Correctness: The claims and methodology parts are technically sound.

Clarity: The structure of the paper is explicit, which is easy to follow and review.

Relation to Prior Work: It clearly clarifies and discusses the technical contributions.

Reproducibility: Yes

Additional Feedback:


Review 3

Summary and Contributions: The authors propose robust recursive partitioning (R2P), a recursive partitioning scheme which incorporates conformal prediction. Instead of a traditional homogeneity criterion (such as mean squared-error), R2P splits to optimize a "confident homogeneity" criterion which penalizes cases where the fixed subgroup means lie outside of covariate-dependent confidence intervals estimated by conformal prediction. This loss is regularized by the subgroup confidence interval widths, to avoid degenerate solutions. R2P is straightforwardly extended from the regression setting into the causal inference setting for estimation of individual treatment effects. Experiments are run on two simulated datasets, as well as the IHDP and CCP semi-synthetic datasets, showing improved across-partition heterogeneity and within-partition homogeneity. --- Update: I thank the authors for their response. In particular I am grateful for helping me understand why they did not evaluate R2P as an ITE estimator, and including Table R1 in the rebuttal. I am increasing my score from a 5 to a 6. My main concern remains the same: "it's difficult to disentangle the effects of a more powerful ITE estimator (the causal GP) from the added benefits of R2P". The authors acknowledge that this is a major drawback of the paper. I would insist they in run appropriate ablation studies for future revisions.

Strengths: The proposed R2P scheme incorporating conformal prediction is quite reasonable and well-motivated. As far as I can tell, this is a novel contribution and likely of relevance to the community.

Weaknesses: The experiments section feels a bit rushed and could use improvement. First, I'd like to see details added to the main text: (1) Explanations of how synthetic datasets A and B were derived (this is already in the Appendix). In particular it's impossible to interpret Figure 3 without details on the data-generating process for synthetic dataset B. (2) Explanations of how hyperparameters (lambda especially) were chosen (some of this is already in the Appendix). The experiments showing the effect of tuning lambda are helpful, but it looks like lambda was fixed to 0 for CPP and 0.5 for all other datasets. What's behind this choice of lambda? If the reader was interested in tuning lambda, what would the authors recommend as a selection criterion? (3) Evaluation on effectiveness of predicting the ITE, for example via RMSE of the estimated ITEs such as in Johansson, Shalit, Sontag ICML 2016. While evaluation on variance across groups and variance within groups is intuitive and valuable, to me RMSE of estimated ITE is a more straightforward way to evaluate estimator effectiveness. (4) Can we add for comparison a baseline an ITE method that's more powerful than simple decision trees? For example, we take predictions from a causal forest and divide subgroups by quantiles of predicted ITE. At the moment all baselines use fairly simple decision tree estimators. As a result it's difficult to disentangle the effects of a more powerful ITE estimator (the causal GP) from the added benefits of R2P. It'd be interesting as well to see R2P applied with a decision tree ITE estimator instead of the causal GP estimator. (5) Table 2 is a bit confusing to me, since R2P isn't compared to any other methods. Would it make sense to compare to, again, subgroups defined by quantiles of predicted ITE as determined by a baseline estimator such as a causal forest? Second, it's difficult to see how the experimental results show that R2P prevents false discovery. This may be implied by some of the differences in variance across subgroups and variance within subgroups, but it's not obvious to me and some explanation would be much appreciated. Is there a more direct way of measuring false discovery? For example a simulation closer to that of Figure 1? Otherwise I'm not sure how much of the claim that R2P prevents false discovery is empirically backed up.

Correctness: For the most part claims and methods look correct to me. See the above section for some discussion on the claim that R2P prevents false discovery.

Clarity: Early parts of the paper are quite well written, especially Sections 1-3. Latter parts feel a bit messier. For example, on lines 284-287, the evaluation metrics of variance are defined in terms of D_l^te. But is this the variance of y for each data point? Or variance of tau? The text specifies the latter, but the notation could be more explicit.

Relation to Prior Work: Relation to prior work is clearly discussed in Section 4.

Reproducibility: Yes

Additional Feedback: On lines 200-201: "Note that confident homogeneity does not necessarily. improve as the group size shrinks because smaller groups lead to greater uncertainty". Could you explain this claim a bit further? It's not obvious to me that a smaller group will always yield wider confidence intervals. Typo on line 271: "here abbreviate". Figure 3: I assume these results are for only one simulation of dataset B. Is there a way to compare the treatment effects across identified subgroups for all 50 simulations?

[Author Response · NeurIPS 2020]

We thank all reviewers for your insightful and constructive comments. Please see our responses below.

**[Reviewer #1]** Please see our responses to your main questions below.

• *Well-identified subgroups*: Theorem 2 guarantees that confidence intervals (CIs) will achieve the required finite
sample coverage for the ITE estimates in each subgroup, regardless of how accurate or inaccurate the underlying ITE
model is. If the CIs exhibit large overlap across the constructed subgroups (because, for example, the ITE estimator
was inaccurate due to covariate imbalance), we can conclude that the constructed subgroups are not well-identified.
Conversely, if the CIs have little or no overlap across subgroups, we can conclude that the subgroups are well-identified.
Given the theoretical guarantee of the CIs in R2P, the subgroups are robust if well-identified. This is an important
advantage of R2P, which we will highlight more clearly in the revised paper.

• *Why not relying on a specific ITE estimator is better*: In recent years, many ITE estimators have been proposed.
However, no one ITE estimator is consistently the best in all settings. Furthermore, these ITE estimators are non-
interpretable black-box models. One of the main contributions of R2P is that it divides units into subgroups with respect
to an interpretable tree-structure, and provides subgroup coverage guarantees for the ITE estimates in each subgroup.
R2P can be combined with any existing ITE estimator, enabling it to play a vital role in producing trustworthy and
interpretable ITE estimates in practice.

**[Reviewer #2]** R2P produces confidence intervals that achieve the required coverage guarantee for the ITE estimates
in each subgroup with respect to an interpretable tree-structure (Theorem 2). This provides upper and lower bounds for
the ITE within each subgroup. Taken together, the coverage guarantee and interpretability of R2P can guide the user to
develop more effective interventions and/or improve the design of further experiments. We will demonstrate this point
clearly in the revised paper, leveraging the superior empirical results of R2P compared to previous methods.

**[Reviewer #3]** We have summarized your main questions and provided our responses below.

• *Baselines with powerful ITE estimators*: Grouping the units based on the quantiles of the estimated ITEs fails
to satisfy the essential requirement of subgroup analysis: interpretability. The estimates from a black-box ITE
estimator are non-interpretable. Similarly, the subgroups defined based on the estimated quantiles give no ex-
planation (in terms of input covariates) regarding why the units are assigned to a particular subgroup. Previous
state-of-the-art subgroup analysis methods are all interpretable but are tied to one particular estimator: decision
tree. While compatible with any black-box ITE estimators, R2P constructs easy-to-interpret subgroups based on
the tree-structure and partition rules of the covariates. In addition, the confidence intervals of R2P achieve cov-
erage guarantees for the ITE estimates in each subgroup. We agree that the performance improvement of R2P
comes both from a better way to construct subgroups *and* the use of a better estimator. However, this is one
of the key advantages of R2P: it is able to use *any* ITE estimator. In the revised paper, we will replicate the
same experiment for R2P using *different* ITE estimators; this should give insight into the source of gain.

• *Table 2 of the paper, "Normalized $V^{in}$"*: In reporting normalized
comparisons across methods, we normalize $V^{in}$ by dividing by $V^{pop}$, the
variance within the entire population. Because the normalizer $V^{pop}$ is the
same for all methods and R2P achieves the smallest $V^{in}$ (Table 1 of the
paper), R2P also achieves the best normalized $V^{in}$ (Table 2 of the paper).
We will clarify this in the revised paper.

Table R1: Average overlap across sub-
groups on Synthetic dataset B.

| R2P | CCT | CT-A | CT-H | CT-L |
|---|---|---|---|---|
| 0.14±.03 | 0.63±.15 | 0.44±.09 | 0.60±.16 | 2.27±.55 |

• *False Discovery*: There is no perfect performance metric for subgroup analysis. The optimal ground-truth of
subgroups depends on multiple objectives, including homogeneity, heterogeneity, and the number of subgroups. In the
literature, the usual metric used is variance, rather than ground-truth, because greater heterogeneity across subgroups
and homogeneity within each subgroup generally imply well-discriminated subgroups. As one metric for evaluating
false discovery, we can use the overlap of treatment effects across subgroups, as in Fig. 3 of the paper. For this, we
suggest *average overlap of treatment effects across subgroups* over 50 simulations. Table R1 here shows that R2P
performs best for Synthetic Dataset B. We would also like to direct you to our response to Reviewer 1 (well-identified
subgroups), in which we highlight how the confidence intervals in R2P can help avoid false discovery.

• *Questions (1-3)*: **(1)** We will add brief explanations regarding datasets A and B in the main text, as per the reviewer's
suggestions. **(2)** The hyperparameter $\lambda$ balances the impact of homogeneity and the width of the confidence intervals,
while $\gamma$ controls regularization. (Experiments in the Supplementary Material demonstrate the impact of $\lambda$ and $\gamma$.) The
choice of $\lambda$ should be made according to the user's prioritization of performance metrics (e.g., $V^{in}$, $V^{across}$, and the
width of confidence intervals). Alternatively, given performance metrics, both $\lambda$ and $\gamma$ can be tuned via cross-validation.
**(3)** RMSE is the appropriate metric to evaluate an ITE estimator. However, R2P is not a method for ITE estimation and
should not be evaluated on that basis; it is a method for subgroup analysis and should be evaluated as such.

• *Additional feedback*: (*Question on lines 200-201*) Smaller subgroups mean smaller sample sizes; smaller sample
sizes lead to wider confidence intervals (for example, the confidence interval for a sample of size 1 would be infinite).
Treatment effects across identified subgroups can be compared via $V^{across}$. (*Question on Figure 3*) The average overlap
in Table R1 shows that the subgroups identified by R2P are well-discriminated over 50 simulations.

[Meta-Review · NeurIPS 2020]

The paper presents a useful and strongly grounded method for identifying heterogeneously responding subgroups in the context of causal treatment effects. The reviewers and I find this method to be novel, and believe it will be of interest to the causal inference community. While many method exists for identifying subgroups for treatment effects, the proposed method can work with black box predictors, has good theoretical guarantees, and focuses on interpretability (which is important in this task). I strongly encourage the authors to conduct the added experiments proposed by the reviewers and incorporate them into the revised version (possibly in the supplement).